# Study and Neural Network Analysis on Durability of Basalt Fibre Concrete

Shanqing Shao [1], Ran Wang [1], Aimin Gong [1], Ruijun Li [2], Jing Xu [3], Fulai Wang [1,*] and Feipeng Liu [3,4,*]

1. College of Water Conservancy, Yunnan Agricultural University, Kunming 650201, China
2. Hubei Academy of Water Resources and Hydropower Sciences, No. 286 Luoshi South Road, Hongshan District, Wuhan 430070, China
3. Institute of International Rivers and Eco-Security, Yunnan University, Kunming 650500, China
4. Southwest Survey and Planning Institute of National Forestry and Grassland Administration, Kunming 650031, China
* Correspondence: ynau115@126.com (F.W.); lfp881214@126.com (F.L.)

**Abstract:** In order to investigate the law of basalt fibre to enhance the durability of concrete, this paper selects basalt fibre length as the main factor, supplemented by novel research methods such as neural networks, to study the rule of concrete resistance to multiple types of salt erosion. Tests have shown that large doses of mineral admixtures and basalt fibres can prolong the time that concrete is eroded by salt solutions; the age of maintenance has a small effect on the mechanical and durability of the concrete; the increase in length of basalt fibres enhances the mechanical properties of the concrete, but weakens the durability. This is exacerbated by the mixing of fibres, but the increase is not significant; the effect of length on concrete resistance to mass loss, corrosion resistance factor of compressive strength, and resistance to chloride ion attack is ranked as follows: 6 mm > 12 mm > 18 mm > 6 mm + 12 mm > 6 mm + 12 mm + 18 mm. The opposite is true for effective porosity; the highest compressive strength corrosion resistance coefficient was found in the length of 6 mm, with an average increase of 6.2% compared to 18 mm, and the mixed group was generally smaller than the single mixed group. The average increase in chloride content was 25.1% for length 18 mm compared to 6 mm; the triple-doped L6-12-18 group was the largest, with an average increase of 33.9% in effective porosity over the minimum 6 mm group. Based on the data from the above indoor trials, artificial neural network models and grey cluster analysis were used to predict and analyse the data, and the prediction and categorisation results were accurate and reliable, providing a reference for subsequent studies.

**Keywords:** basalt fibres; durability; artificial neural networks; grey cluster analysis





## 1. Introduction

In more than 40 years of reform and opening up, China's economy has developed extremely rapidly. Concrete buildings have evolved with the times, with a gradual increase in large projects such as large span bridges and heavily laden hydraulic buildings. These buildings have high requirements for a range of mechanical properties, workability and durability of concrete [1]. Considering the environment in which such buildings are located, ordinary conventional concrete, with its disadvantages such as brittleness, susceptibility to cracking, poor durability and serious cement consumption [2], restricts its use.

There was a major construction effort in the West, with the building of a large number of concrete infrastructure facilities. Additionally, the soil in the area contains high levels of corrosive salts that can cause corrosion to the concrete; among the main corrosive substances are $Cl^-$, $SO_4^{2-}$ etc. [3]. $SO_4^{2-}$ reacts with concrete, and the resulting calcium alumina and gypsum are expansive. Pre-corrosion is mainly manifested by the precipitation of crystals on the concrete surface, and the late stage of corrosion by spalling in the area of concrete adsorption and exposed aggregates [4]. $Cl^-$ can cause damage to the passivation

film of reinforcing steel in concrete, accelerating local corrosion of reinforcing steel, causing expansion and cracking of the concrete protection layer [5]. Salt corrosion poses a serious threat to concrete buildings, significantly shortening their service life and causing them to deteriorate at an accelerated rate. Therefore, research into the durability of concrete against salt erosion in coastal and western areas is essential.

In the face of the significant increase in concrete buildings, the consumption of concrete raw materials is increasing day by day. This will determine the development of concrete in the direction of green materials, reduced consumption of raw materials, high quality, high durability and environmental protection. Basalt fibres have the advantages of stable chemical properties, good thermal shock stability and high tensile strength, being green non-polluting, and corrosion-resistant, etc. Basalt fibre fly-ash concretewhich has the advantages of both basalt fibre concrete and fly-ash concrete, was then synthesised. The mixing of basalt fibres with fly-ash has a synergistic effect in concrete, contributing to the improvement of concrete in terms of compactness, tensile strength and erosion resistance, improving the mechanical properties and durability of concrete. Lilli Matteo [6] et al. used plasma-enhanced basalt fibres to deposit vinyl silane on the surface of basalt fibres. The results show that the shear strength of the modified basalt fibre concrete increased by 79%, highlighting the excellent properties of the composite. Venkat Raman et al. [7] studied the mechanism of basalt fibre-reinforced concrete and tested the flexural performance of concrete; they found that basalt fibre-wrapped concrete has higher flexural performance, and greater load bearing capacity and economy compared to conventional concrete. Meyyappan PL [8] et al. mixed basalt fibres (0.5%, 1%, 1.5%, 2%, 2.5% and 3%) into concrete for the purpose of enhancing the tensile strength of concrete, and the strength performance of concrete was optimal when the volume dose was 1%. The strength properties tend to decrease with further increases in volume dosing. Sadjad Pirmohammad [9] studied the effect of basalt fibres of different admixtures (0.1%, 0.2%, 0.3%) and different lengths (4 mm, 8 mm, 12 mm) on concrete, using fracture tests to standardise the degree of correlation. The results show that the fracture toughness increases with the increase of fibre content; the fracture toughness decreases with the increase of basalt fibre length; the fracture toughness of basalt fibre concrete with 0.3% volume admixture and 4 mm length is the highest. Zhi Pin Loh [10] used two different fibres of polyvinyl alcohol (Penicillin V Acylase) and basalt to enhance the performance of concrete, and found that the fibres had little effect on the compressive strength of FRCC, but there was a significant increase in splitting tensile and flexural strength. Nasir Shafiq [11] investigated the compressive strength of short-cut basalt fibre concrete, with the volume admixture, using the non-destructive testing technique ultrasonic pulse velocity (UPV). Luigi Fenu [12] discusses the laws of basalt fibre-reinforced concrete, analysing the energy absorption and tensile strength of concrete at high strain rates. The results show that the addition of basalt fibre has a non-significant effect on the dynamic increase factor and only slightly increases it. El-Gelani A. M. et al. [13], investigated the effect of basalt fibre-reinforced concrete on mechanical properties. It was found that the relative modulus of elasticity, splitting tensile strength, and flexural strength of the concrete was significantly enhanced with the incorporation of basalt fibres, while the compressive strength was slightly improved. Therefore, under the premise of green, environmental protection and economy, the green building materials fly-ash and basalt fibre were introduced into the development of concrete with the aim of improving the mechanical properties and durability of concrete; this has carried on the correlation research. In light of this, achieving a comprehensive grasp of basalt fibre fly-ash concrete to provide a realistic basis for its application in the field of civil engineering has important significance.

The artificial neural network approach simulates the structure of neurons in the human brain, which receives and processes information from outside. Artificial neural network techniques are now used more and more maturely, and are also used more and more in engineering [14–16]. The main methods of machine learning currently applied to engineering are algorithms such as artificial neural networks, random forests, support vector machines and decision trees. Where artificial neural networks have a huge advantage

is the reflection of the uncertain relationship between the independent and dependent variables; artificial neural networks can make non-linear connections between inputs and outputs [17–19], and accurate and efficient models can therefore be built quickly. The second advantage is these algorithms' ability to learn. Artificial neural networks can use several mathematical algorithms simultaneously to create weights between neurons efficiently, and this process of creating and determining weights is an optimisation method that reflects the powerful learning ability of artificial neural networks.

Among them, the use of artificial neural networks to predict the compressive strength of concrete is more studied. For example, Garg A, Aggarwal P, Aggarwal Y, et al. [20], using SVM and GPR to predict the compressive strength of concrete containing silica nanoparticles, found results that show that SVM performs better in predicting the results. Kandiri Amirreza [21] predicted the compressive strength of concrete containing recycled aggregates and tested the validity of the algorithm using the M5P tree model; Ahmad Ayaz [22] used supervised machine learning (ML) algorithms, gene expression programming (GEP) and artificial neural networks (ANN) to predict the compressive strength of rca-based concrete in comparison. The GEP model had a correlation (R2) value of 0.95, while the ANN model had an R2 value of 0.92, making the GEP model more effective in prediction; Adriana Trocoli Abdon Dantas [23] used an artificial neural network model to predict the compressive strength of concrete containing construction and demolition waste (CDW) at 3, 7, 28 and 91 days, respectively. Hosein Naderpour [24] employed a model based on a neuro-fuzzy approach as a soft computing technique, and this model predicted the compressive strength of environmentally friendly concrete more accurately; Several scholars have used artificial neural networks to predict indicators of chloride ion erosion in concrete, and Taffese Woubishet Zewdu [25] used the XGBoost machine learning algorithm to predict the migration coefficient of chloride ions in concrete; Sun et al. [26] used a multi-scale simulation approach and employed a spherical inclusions model to calculate the diffusion coefficient of chloride ions in concrete.

Research on concrete durability using artificial neural networks has gradually started to develop in recent years [27–31]. Liu Kaihua [32] used a machine learning model to predict the carbonation depth of recycled aggregate concrete and showed that the random forest model had better performance than the Gaussian progression regression model and the independent artificial neural network (ANN) model. Z.H. Duan [33] investigated the applicability of artificial neural networks (ANN) in modelling the modulus of elasticity (Ec) of recycled aggregate concrete (RAC), and the results of the study showed that the established neural network model was better in predicting the modulus of elasticity of concrete from different sources of RA materials. Most of these are predictive models for carbonation, porosity and resistivity. However, there has been relatively little research into the durability of fibre fly-ash concrete.

This paper investigates the mechanism of resistance to multi-salt erosion under each factor by studying the effect of basalt fibre length, curing age and other factors on mass loss rate, compressive strength corrosion resistance factor, effective porosity and chloride ion content in basalt fibre fly-ash concrete. Eight algorithms for artificial neural networks were written using a python program to find the best-fit model for basalt fibre fly-ash concrete durability prediction by seeking prediction results with higher prediction scores. A clustering analysis was carried out on basalt fibre fly-ash concrete's durability related indicators, based on Matlab software using grey clustering analysis.

## 2. Materials and Methods

### 2.1. Materials

P.O42.5 ordinary Portland cement was selected for cement, and its chemical properties meet national standards, and the physical and mechanical properties are shown in Table 1.

**Table 1.** Cement properties.

| Compressive Strength/MPa | | Flexural Strength/MPa | | Condensation Time/min | |
|---|---|---|---|---|---|
| 3 d | 28 d | 3 d | 28 d | Initial coagulation | Final coagulation |
| 21.4 | 46.8 | 6.2 | 8.6 | 155 | 230 |

Basalt fibre is selected from basalt chopped fibre, and its main physical and mechanical performance indicators are shown in Table 2.

**Table 2.** Physical and mechanical properties of basalt fibres.

| Length (mm) | Diameter (μm) | Density (g/cm$^3$) | Elastic Modulus (GPa) | Tensile Strength (MPa) |
|---|---|---|---|---|
| 6/12/18 | 15 | 2.65 | 91 | 3900 |

The coarse aggregate adopts continuous graded crushed stone with a particle size of 5~20 mm; The fine aggregate adopts machined sand with a fineness modulus of 2.6 and an apparent density of 2670 kg/m$^3$.

Fly ash is selected secondary fly ash, and its main chemical composition and physical properties are shown in Tables 3 and 4 respectively.

**Table 3.** Chemical composition of grade ii fly ash/%.

| Chemical Composition | SO$_3$ | CaO | SiO$_2$ | Al$_2$O$_3$ | Fe$_2$O$_3$ | MgO |
|---|---|---|---|---|---|---|
| Content | 0.36 | 2.83 | 50.26 | 31.14 | 4.16 | 0.78 |

**Table 4.** Physical properties of fly ash.

| Apparent Density (g/cm$^3$) | Moisture Content (%) | Strength Activity Index (%) | Loss on Ignition (%) | Fineness (%) | Water Demand Ratio (%) |
|---|---|---|---|---|---|
| 2.25 | 0.52 | 86 | 2.34 | 20 | 98 |

*2.2. Test Mixes*

In order to investigate the basic mechanical properties of basalt fibre fly-ash concrete, the test mix was designed with a strength of C40, a water-cement ratio of 0.36 and a sand ration of 40%. Of all the raw materials, aggregates make up the majority of the concrete, and the moisture they contain has an important influence on the concrete mix design. It is therefore important to carry out sand and stone moisture content tests to control the moisture content in aggregates. The preliminary mix ratio is converted into a test mix ratio by adjusting the influencing parameters according to the actual moisture content and strength of the aggregates used. The specific parameters are shown in Table 5.

**Table 5.** Mix proportion concrete (kg/m$^3$).

| Number | Cement | Fly Ash | Mineral Powder | Fine Aggregates | Coarse Aggregates | Water | Water Reducing Agents | Short-Cut Basalt Fibres (%) | | |
|---|---|---|---|---|---|---|---|---|---|---|
| | (kg/m$^3$) | | | | | | (%) | 6 mm | 12 mm | 18 mm |
| $C_{0.12}L_6$ | 237 | 118.5 | 39.5 | 735 | 1089 | 142 | 0.13 | 0.12 | — | — |
| $C_{0.12}L_{12}$ | 237 | 118.5 | 39.5 | 735 | 1089 | 142 | 0.27 | — | 0.12 | — |
| $C_{0.12}L_{18}$ | 237 | 118.5 | 39.5 | 735 | 1089 | 142 | 0.30 | — | — | 0.12 |
| $C_{0.12}L_{6-12}$ | 237 | 118.5 | 39.5 | 735 | 1089 | 142 | 0.29 | 0.06 | 0.06 | — |
| $C_{0.12}L_{6-12-18}$ | 237 | 118.5 | 39.5 | 735 | 1089 | 142 | 0.29 | 0.04 | 0.04 | 0.04 |

Note: Where $L_6$, $L_{12}$, $L_{18}$, $L_{6-12}$, $L_{6-12-18}$ represent short-cut basalt fibre lengths of 6 mm, 12 mm, 18 mm, 6 mm mixed with 12 mm (1:1) and three lengths mixed (1:1:1) respectively.

*2.3. Durability Test Methods*

The durability test methods for basalt fibres are shown in Table 6.

**Table 6.** Durability test methods.

| Content of the Test | Specific Test Indicators |
|---|---|
| Conservation age | 28 d, 56 d, 84 d |
| Erosion solutions | Sodium sulphate at 10%, sodium chloride at 7% |
| Erosion patterns | Soaking erosion |
| Fibre length | 6 mm, 12 mm, 18 mm, 6 mm + 12 mm, 6 mm + 12 mm + 18 mm |
| Fibre bulk rate | 0.12% |
| Erosion time | 0d, 30 d, 60 d, 90 d, 120 d, 180 d, 240 d |
| Test content | Compressive strength corrosion resistance factor, mass loss rate, effective porosity, chloride ion content |

2.3.1. Mechanical Properties Test Procedure

Basalt fibre fly ash concrete specimens were tested for cubic compressive and splitting tensile strengths after reaching the design curing age by removing and drying the surface water. Its mechanical properties were tested in strict accordance with the specifications.

2.3.2. Durability Test Procedure

(1)  The 100 mm × 100 mm × 100 mm cube specimen was taken out after 26 d of standard maintenance, wiped off the surface moisture and placed in a ventilated environment for 48 h to dry out the moisture.

(2)  The dried specimens were placed in a well-configured mixed solution tank; the solution should be 20 mm above the highest layer of concrete specimens and the spacing between specimens should not be less than 20 mm. The immersion time was calculated when the solution is ready, and the solution needs to be prepared for less than 30 min and replaced every 30 d. The solution should be temperature controlled to avoid errors caused by temperature, which should be controlled at 20 °C ± 2 °C.

(3)  When the concrete specimens are removed after erosion to the specified age for the compressive test, they should be wrapped in wet towels and transported to the laboratory for the compressive test; to carry out the mass loss test, a balance must first be prepared with an accuracy of 0.01 g, then the surface of the specimen should be dried, removed from the erosion tank and immediately weigh the saturated mass of concrete; to carry out the porosity test, the weighed saturated concrete specimens were wrapped in towels, transported to the laboratory, dried in an oven at a controlled temperature of 105 °C ± 5 °C, dried at high temperature for 48 h, then weighed again for mass and finally the corresponding chloride ion measurements.

*2.4. Artificial Neural Networks*

The artificial neural network approach simulates the structure of neurons in the human brain, which receives and processes external information, thus achieving an intelligent solution function similar to that of the human brain. It is the model that plays an important role in the activity of artificial neural networks [34]. The basic building blocks of the human brain are neurons, which are capable of thinking and learning from experience and remembering. Many parallel structures make up the artificial neural network; these cells receive data in the form of weighted inputs and then activate a series of neurons by transferring the weighted outputs to other neurons through an activation function.

ANN is one of the most advanced artificial intelligence technologies available and it has many advantages. One is a reflection of the uncertain relationship due to the dependent and independent variables. The ANN method is able to achieve a non-linear connection between input and output due to its excellent mathematical tools. This allows a valid predictive model to be built up quickly. In addition, ANN has the advantage of improving learning ability. ANN can effectively establish the weights between neurons using multiple mathematical algorithms simultaneously. This process of creating and determining weights is an optimised approach that reflects the powerful learning capabilities of artificial neural networks. At the same time, the algorithm has some drawbacks, such as the large amount of data needed to train the algorithm and the difficulty in explaining the internal mechanism and the memory.

Training data gives good results, while test data produces bad results, the so-called memory. To prevent this problem, the optimal model needs to be tested by examining the degree of error in the training, validation and test data.

The input, hidden and output layers is the basic structure of the ANN model. ANN models all have at least one input layer, a hidden layer and an output layer, and there can be more than one hidden layer. For very complex problems, multiple hidden layers can be very helpful. The ANN model is made up of the neurons, as shown in Figure 1. Although only the values of the independent variables are present in the input layer neurons, mathematical operations are performed in the hidden and output layers. Equation (1) represents the mathematical operations performed in the neuron.

$$y = f \left( \sum wi\ xi + b \right) \tag{1}$$

$y$-represents the result of the neuron,
$w$-represents the inter-neuron weights,
$x$-means input from the upper level,
$f$-stands for transfer function,
$b$-represents deviation,
$i$-represents the number of neurons.

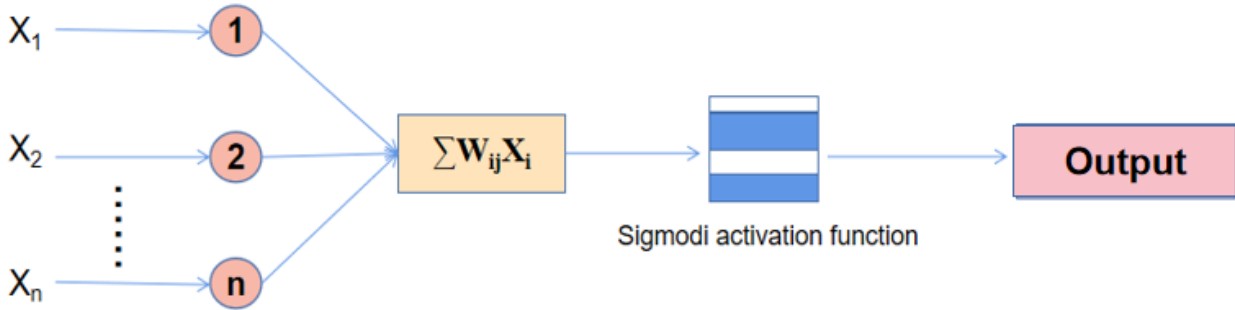

**Figure 1.** Demonstration of neuronal model.

The input obtained from the previous layer is multiplied by the weighted and increasing deviations. The outcome of the neuron depends on the processing of this accumulation

of transfer functions. This result is multiplied by the weights and dispersed to the next layer. The neural network architecture consists of an input layer, a hidden layer and an output layer, as well as other elements. At each layer, with the exception of the input layer, linear combinations with biases are analysed first.

*2.5. Decision Trees*

The decision tree algorithm is a computational method to approximate the value of a discrete function. This is a typical classification method which works by processing the data and then generating readable rules and decision trees through inductive operations, and finally analysing the new data through decision making. Essentially, a decision tree is a set of rules used for classifying. The decision tree algorithm discovers the classification rules hidden in the data by constructing a decision tree. The core component of the decision tree algorithm is the construction of highly accurate, small-scale decision trees. The decision tree is constructed in two stages. The first step is to generate a decision tree: a decision tree is generated from the sample pool of the training set. Typically, training sample datasets are historical data with a certain degree of synthesis, wherein the data are analysed and processed according to actual needs. The second step is decision tree pruning; decision tree pruning is the process of checking, correcting and trimming down the decision tree generated in the previous stage; it focuses on pruning branches that affect prediction by examining the initial rules generated during the generation of the decision tree with data in a new sample dataset (the so-called test data set).

*2.6. Grey Clustering Analysis*

Grey clustering can be divided into two types: one is grey correlation clustering, which is used for the grouping of similar factors, and the other is grey-whitening weight clustering, which is used to detect the class to which an observation belongs.

Using grey correlation clustering analysis, we analyse whether several of the many factors are more highly correlated; we use a composite average indicator of these factors or one of them to represent them without significant loss of information, thus allowing us to save costs and money by reducing the collection of unnecessary variables (factors) through grey correlation clustering of typical sample data before conducting large scale research.

## 3. Results and Discussion

*3.1. Effects of Length on Quality Loss Rate*

Figure 2a,b reflect the variation pattern of basalt fibre fly-ash coagulation mass loss rate with the fibre length. With the increase in erosion time, the concrete mass tends to increase and then decrease, with the mass increasing phase from 0d to 90 d and the mass decreasing phase after 90 d. The erosion time starts to show mass reduction at 240 d; for example: lengths of triple-doped C0.12L6–12-18 [6 mm, 12 mm and 18 mm mixed in three lengths (1:1:1)] underwent a mass loss rate change from negative to positive, that is, a mass reduction. It can be deduced that all other lengths of specimens will also successively show positive values after 240 d, with a loss of mass. As the length of basalt fibre increases, the rate of mass loss also tends to increase, and the rate of mass loss is greater for mixed basalt fibre lengths than for single mixes.

From experimental results, it can be seen that the choice of basalt fibre length has a certain effect on the concrete mass loss rate, but it is not significant. For basalt fibre lengths, the descending order of resistance to mass loss is 6 mm > 12 mm > 18 mm > 6 mm + 12 mm > 6 mm + 12 mm + 18 mm; this is due to the fact that at the same volume, the shorter lengths of basalt fibres are more dense in comparison to the quantity, the concrete fills more densely and binds the concrete surface debris more strongly. For mixed fibres, it is easy to form agglomerates within the concrete; between the mass is a weak link, and the erosion solution finds it easy to penetrate deep into the concrete, accelerating the erosion of the concrete and resulting in accelerated quality loss of concrete.

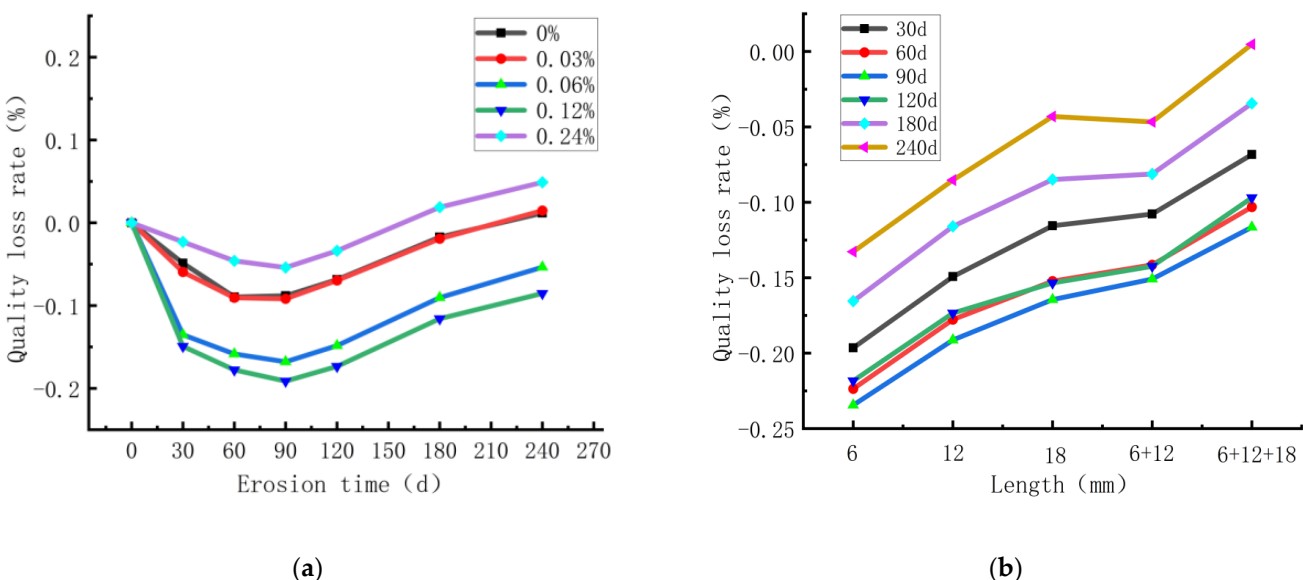

**(a)**                                                                                                                    **(b)**

**Figure 2.** (**a**) Influence of length on mass loss rate (**b**) Influence of length on mass loss rate.

### 3.2. Effects of Length on Effective Porosity

The effective porosity can directly reflect the change in pore filling of concrete specimens. Figure 3a,b show the change in effective porosity with increasing erosion time for different lengths of basalt fibres of the same volume ratio. The effective porosity of basalt fibre fly-ash concrete, with the increase in erosion time, always shows a trend of first declining and then rising. Thirty days is the lowest point of effective porosity, and after 60 d, this grows slowly. The tendency is to increase with the length of basalt fibres, with the fibre mix being significantly larger than the single group. The triple-doped C0.12L6–12–18 group was the largest, with an average increase in effective porosity of 33.9% over the minimum 6 mm group. For basalt fibre lengths, the effective porosity in descending order is: 6 mm + 12 mm + 18 mm > 6 mm + 12 mm > 18 mm > 12 mm > 6 mm.

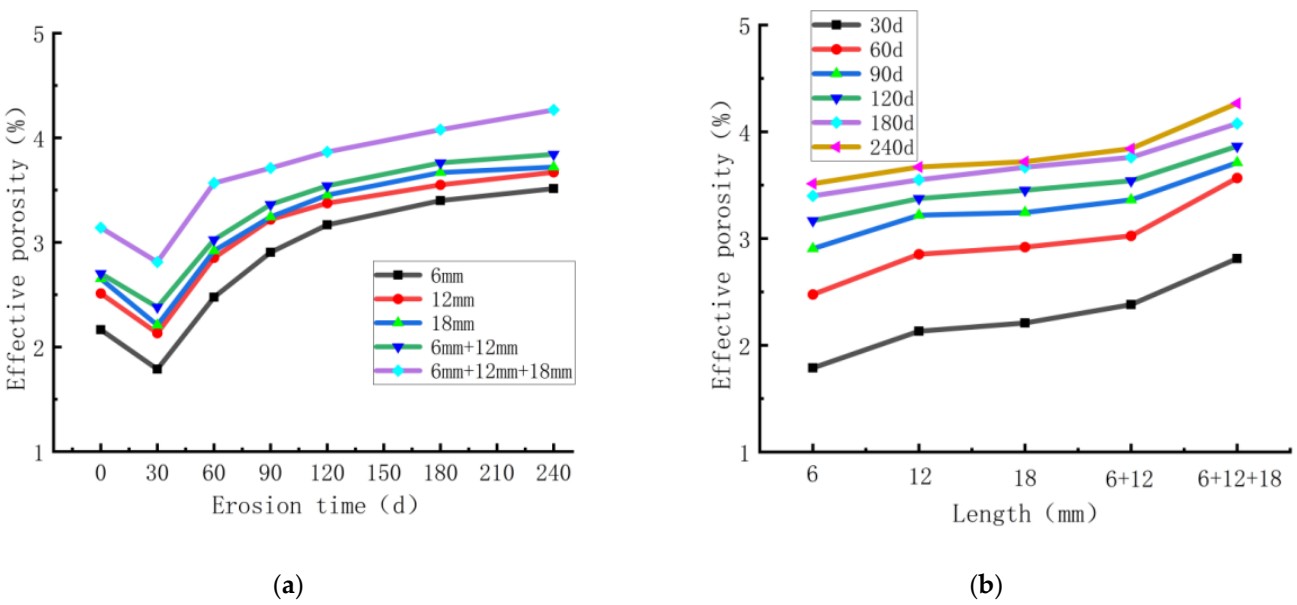

**(a)**                                                                                                                    **(b)**

**Figure 3.** (**a**) Effect of length on effective porosity (**b**) Effect of length on effective porosity.

The reason for this is that the shorter the length of the basalt fibre at the same volume ratio, the greater the relative increase in its order of magnitude, the greater the reduction in pore connectivity within the concrete, resulting in a reduction in porosity. The mixed fibres, on the other hand, are weak due to more agglomeration [35], which leads to more cracks and a larger effective porosity.

### 3.3. Effects of Length on the Corrosion Resistance Factor of Compressive Strength

From Figure 4a,b, it can be seen that the coefficient of compressive strength corrosion resistance of basalt fibre fly-ash concrete tends to rise and then fall with the growth of erosion time, reaching a peak at 90 d and slowly falling after 120 d. As the length increases, it tends to decrease; the highest compressive strength coefficient of corrosion resistance is found in the 6 mm length, with an average increase of 6.2% over the 18 mm length, and the mixed group is generally smaller than the single group. For basalt fibre length, the compressive strength corrosion resistance factor of basalt fibre fly-ash concrete is ranked from largest to smallest as follows: 6 mm > 12 mm > 18 mm > 6 mm + 12 mm + 18 mm > 6 mm + 12 mm.

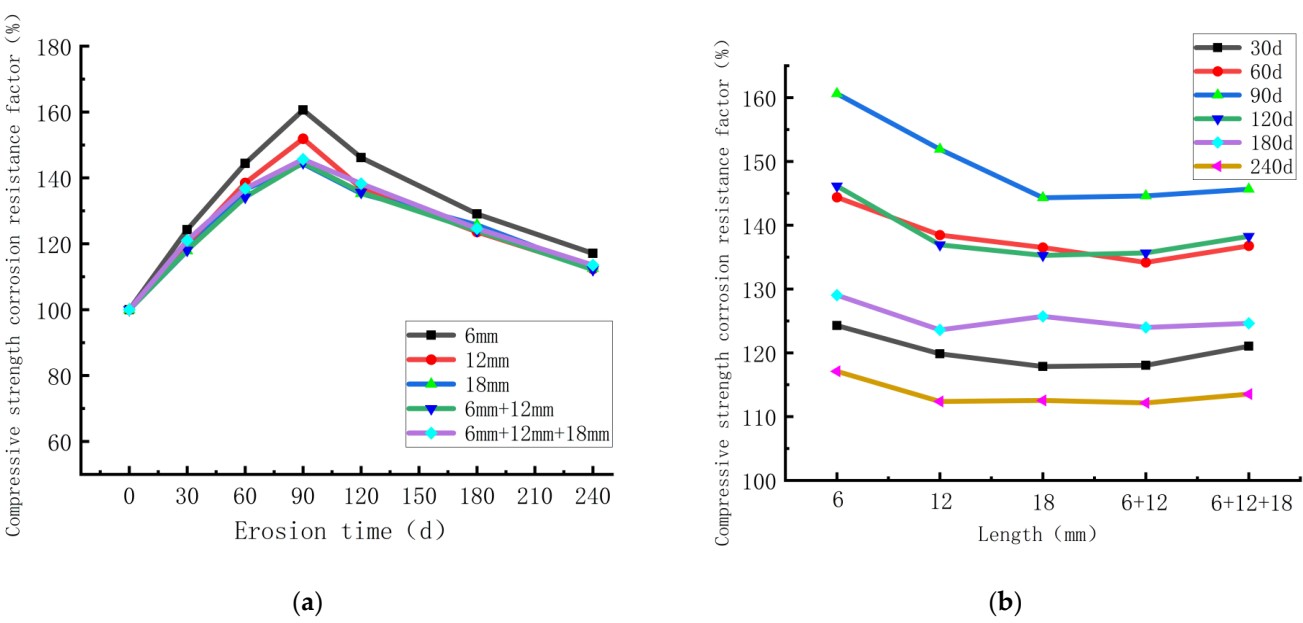

(**a**) (**b**)

**Figure 4.** (**a**,**b**) Influence of length on corrosion resistance coefficient of compressive strength.

Experiments have shown that the length of basalt fibres has a significant effect on the corrosion resistance factor of compressive strength, that shorter fibres are more susceptible to erosion by salt solutions than longer ones, and that with larger fluctuations in strength, the mixed group is less affected by erosion than the single group. This is due to the fact that the longer the fibres, the higher the binding force on the concrete and the better it can counteract the expansion stress caused by mixed salt, reducing the external forces that the concrete needs to resist and causing the corresponding compressive strength corrosion resistance factor to be smoothed out. The mesh structure built up by the mixed basalt fibres is slightly more binding than the single-doped group.

### 3.4. Effects of Length on Chloride Content

Figure 5a,b show the pattern of the effect of basalt fibre length on the chloride ion content after different times. The chloride ion content tested in the trials was all at 2 mm below the surface of the basalt fibre fly-ash concrete. The chloride ion content of basalt fibre fly-ash concrete tends to rise with increasing erosion time. The chloride content rises rapidly until 120 d, after which the rate of increase slows down; the trend increases slowly

with the length of the basalt fibre, with an average increase of 25.1% for lengths of 18 mm compared to 6 mm. At the same time, thte chloride ion content in the mixed group was significantly higher than in the single group. For basalt fibre length, the basalt fibre fly-ash concrete resistance to chloride ions is ranked from largest to smallest as follows: 6 mm > 12 mm > 18 mm > 6 mm + 12 mm > 6 mm + 12 mm + 18 mm.

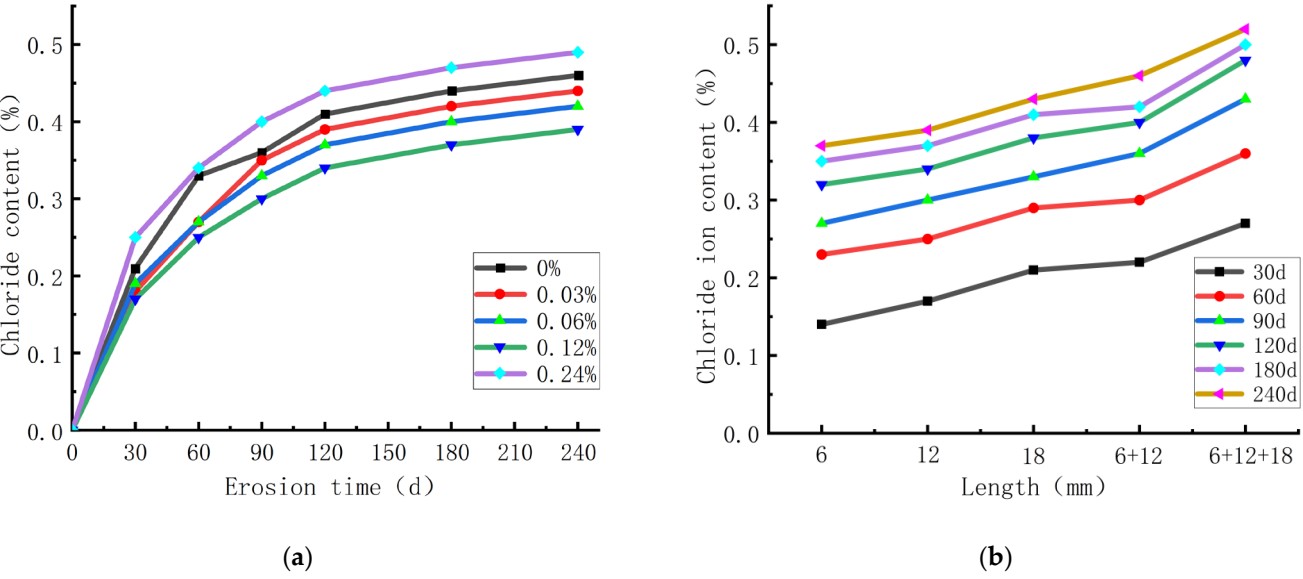

(**a**)　　　　　　　　　　　　　　　　　　　　　　　　　(**b**)

**Figure 5.** (**a**) Influence of length on chloride ion content (**b**) Influence of length on chloride ion content.

Figure 6 shows the chloride ion content at a depth of 0 mm to 30 mm below the surface of the specimen after 90 d of erosion. The pattern of variation in chloride ion content along the depth direction for different lengths of basalt fibres of the same volume ratio is shown. The chloride content tends to decrease with increasing depth. At a depth of 15 mm, the decreasing trend slows down and plateaus.

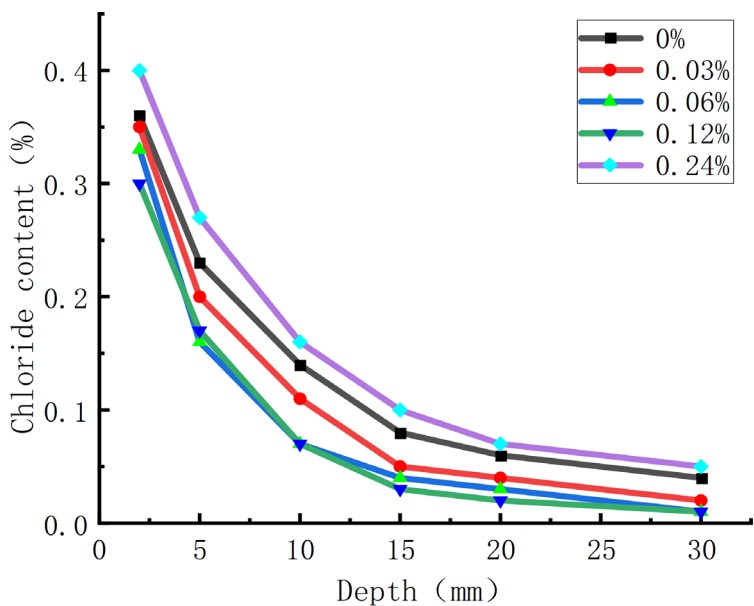

**Figure 6.** Chloride ion content at 90 days of erosion.

This is due to the fact that the increase in length of basalt fibres leads to a decrease in concrete compactness and an increase in porosity, which makes it easier for chloride ions to

enter the concrete, resulting in a higher chloride ion content. Similarly, the mixed group produces more inter-cluster cracks than the single group, and this weakness leads to higher chloride ion content in the mixed group.

### 3.5. Effects of Curing Age on the Resistance of Concrete to Multi-Salt Attack at Different Fibre Lengths

Table 7 shows the results of the concrete specimens tested for each index after 90 days of immersion erosion under different curing ages. Basalt fibre fly-ash concrete, with the growth of curing time, showed a decreasing trend except for the compressive strength; the mass loss rate, effective porosity and chloride content trend upwards with increasing length. The concrete cube compressive strength increases first and then decreases, with the specimens numbered C0.12L6 + 12 lying between C0.12L6 and C0.12L12and the C0.12L6 + 12 + 18 group being the strongest. This is due to the fact that in the single group, the optimum length is 12 mm, and that the concrete with triple-mixing of the basalt fibres has the best load-bearing capacity due to the mesh structure.

**Table 7.** Corrosion resistance test results of different curing ages.

| Length (mm) | | 6 | 12 | 18 | 6 + 12 | 6 + 12 + 18 |
|---|---|---|---|---|---|---|
| Compressive strength (MPa) | 28 d | 72.6 | 74.7 | 73.3 | 72.9 | 75.5 |
| | 56 d | 74.1 | 76.6 | 74.7 | 75.1 | 76.9 |
| | 84 d | 74.6 | 76.9 | 75.4 | 75.8 | 77.4 |
| Quality loss rate (%) | 28 d | −0.2345 | −0.1913 | −0.1645 | −0.1509 | −0.1165 |
| | 56 d | −0.2662 | −0.2271 | −0.2035 | −0.1859 | −0.1541 |
| | 84 d | −0.2737 | −0.2353 | −0.2147 | −0.1982 | −0.1697 |
| Effective porosity (%) | 28 d | 2.91 | 3.22 | 3.24 | 3.36 | 3.71 |
| | 56 d | 2.53 | 2.67 | 2.83 | 2.95 | 3.30 |
| | 84 d | 2.47 | 2.58 | 2.75 | 2.88 | 3.15 |
| Chloride ion content (%) | 28 d | 0.27 | 0.30 | 0.33 | 0.36 | 0.43 |
| | 56 d | 0.27 | 0.28 | 0.31 | 0.33 | 0.39 |
| | 84 d | 0.26 | 0.27 | 0.30 | 0.31 | 0.37 |

### 3.6. Artificial Neural Network Prediction Analysis

In this paper, an artificial neural network model written in python software is used, including eight algorithms of artificial neural network model. The eight algorithms are random forest, decision tree, SVC, KNN, logistic regression, linear SVC, perceptron and stochastic gradient decent. Eight algorithmic artificial neural network models were used to predict and estimate the durability index of basalt fibre fly-ash concrete, and the prediction accuracy of the eight models was compared to select the best model.

The artificial neural network input layer established in this paper has six units, which are the water to cement ratio, fly ash, water reducing agent, basalt fibre length, compressive strength and age. The output layer has three units, namely mass loss rate, effective porosity and chloride content. The prediction results' software scores are shown in Figure 7, from which it is clear that the decision tree artificial neural network prediction results' software scores are the highest.

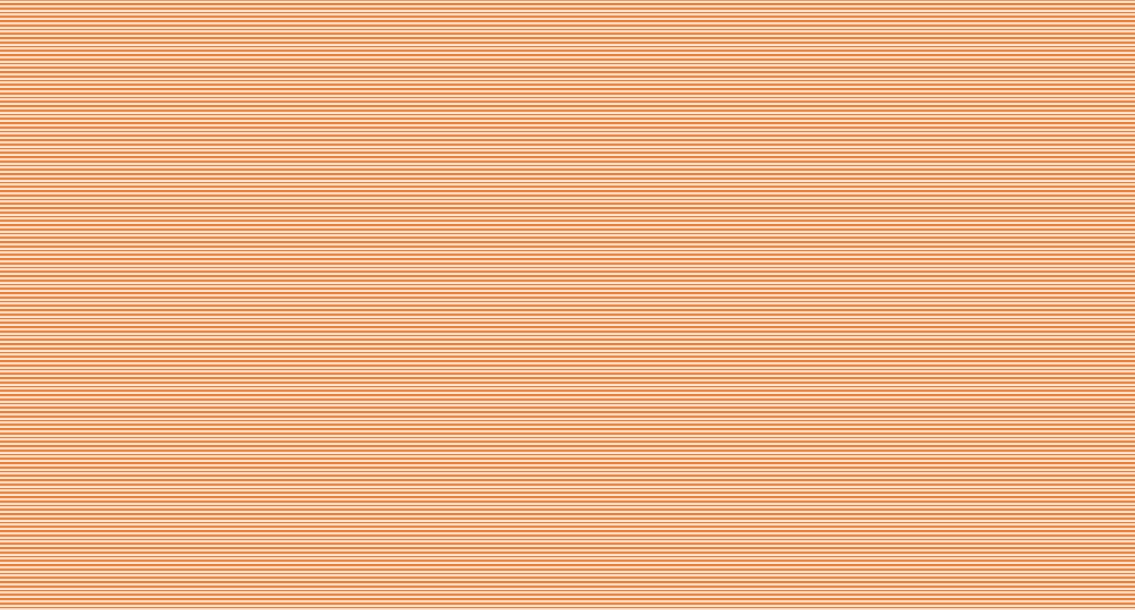

**Figure 7.** Software scoring of each algorithm model.

When using the decision tree algorithm, the machine learning module based on python software predicts the mass loss rate, effective porosity and chloride content of basalt fibre fly-ash concrete; the predicted values closely matched the rate of change of the measured values with very high accuracy, with an absolute error of <1% for all samples and a software score of 99.6 for the prediction results. It shows that the decision tree model has good application. A comparison of the artificial neural network prediction results of the decision tree model with the experimental results is shown in Figures 8–10.

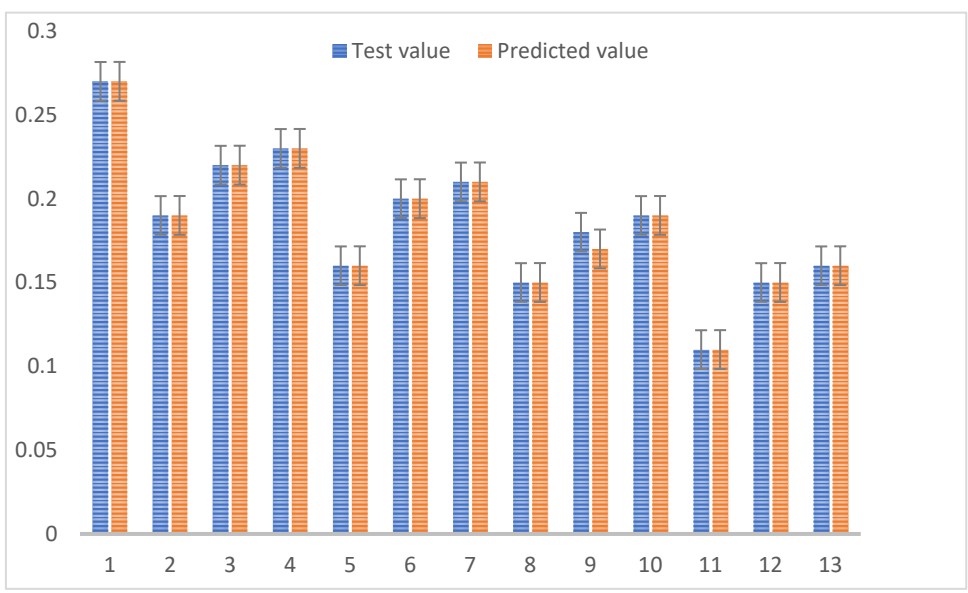

**Figure 8.** Distribution of measured and predicted errors of mass loss rate.

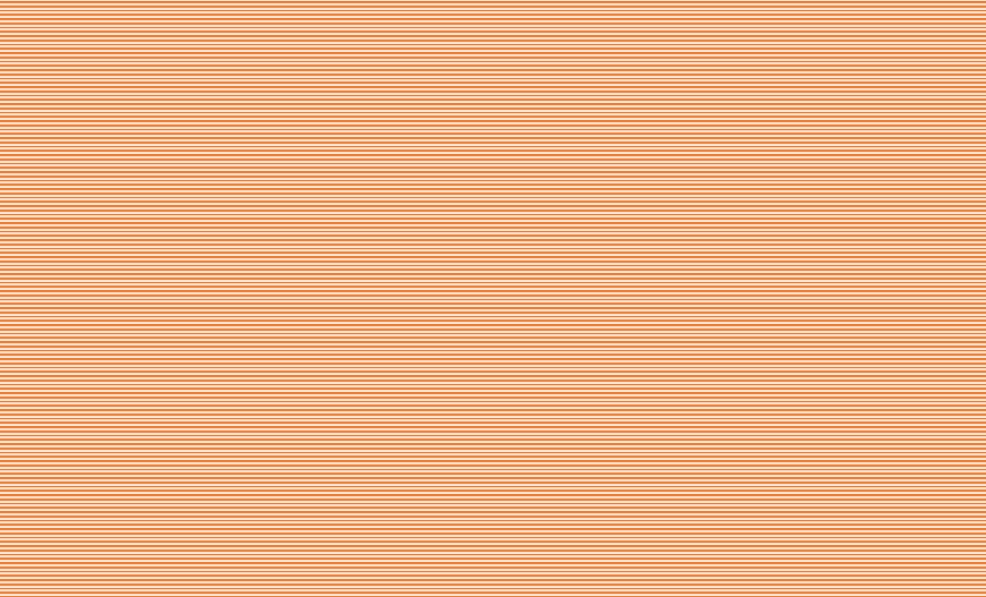

**Figure 9.** Measured and predicted error distribution of effective porosity.

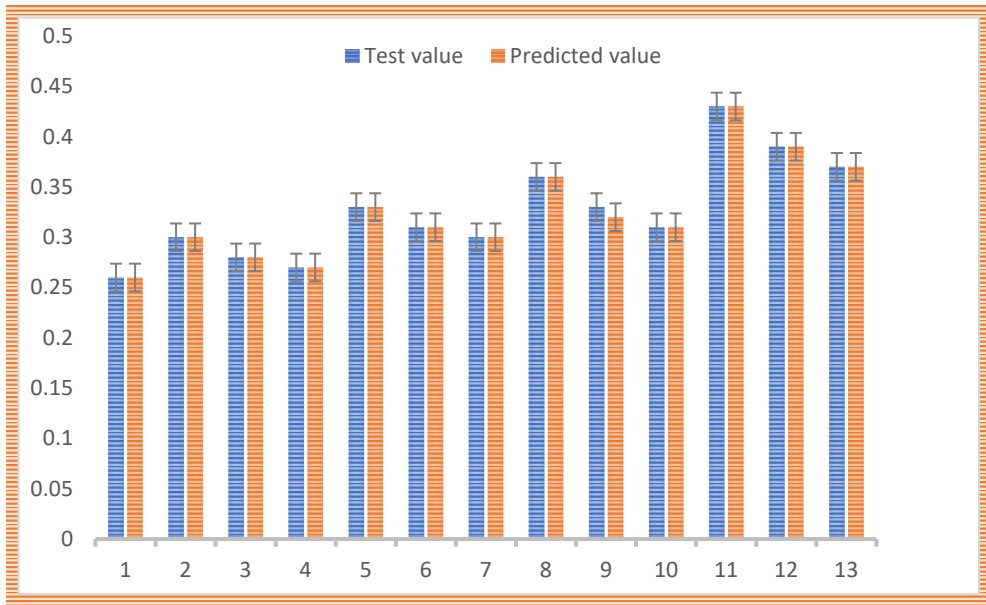

**Figure 10.** Distribution of chloride ion content measurement and prediction error.

### 3.7. Grey Clustering Analysis

Grey clustering is a method of classifying a number of indicators and observations into definable categories based on an association matrix, where a cluster can be seen as a collection of observations belonging to the same category. As the grey composite correlation examines both similarity and proximity between data series, grey correlation clustering can reduce the collection of unnecessary variables (factors) and therefore save costs and money. Therefore, a comprehensive correlation degree model was used to analyse the intrinsic connection between the durability indicators of basalt fibre fly-ash concrete, and the calculation results are shown in Figure 11.

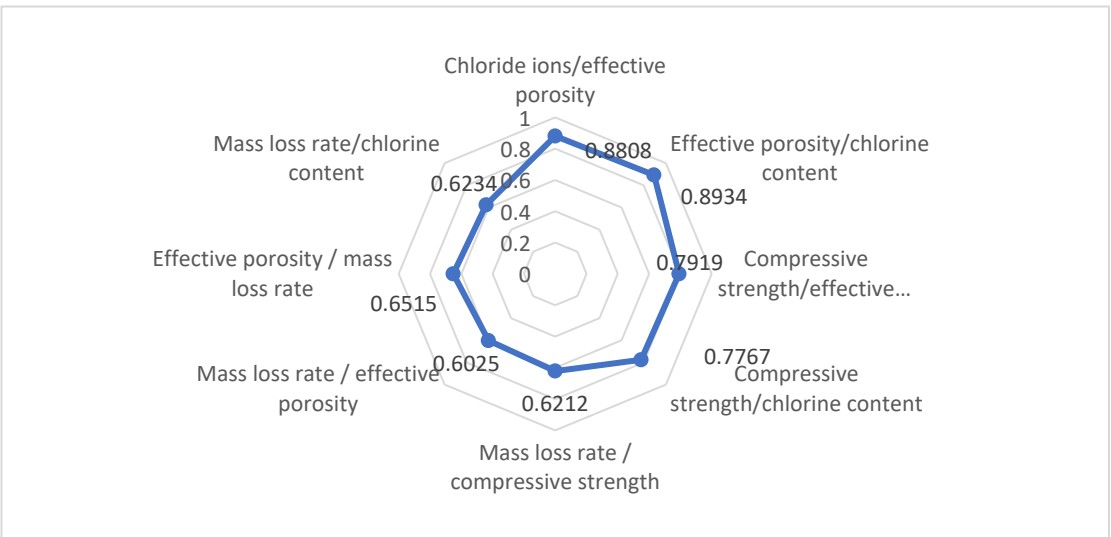

**Figure 11.** Correlation of durability index of basalt fibre fly-ash concrete.

Figure 11 shows that the combined correlation values between compressive strength, effective porosity and chloride content are all close to or greater than 0.8, which is greater than the correlation values between mass change rate and the other performance indicators. The reason for this is that compressive strength, effective porosity and chloride ion content reflect the changes in the internal pore structure of basalt fibre fly-ash concrete from different aspects; the large porosity, loose internal structure, and macroscopic performance of compressive strength is reduced; the effective porosity and chloride ion content are decreased. The rate of change in the quality of basalt fibre fly-ash concrete correlates less well with other macroscopic indicators, indicating that the change in the quality of basalt fibre fly-ash concrete does not reflect well the change in its internal soil pore structure. In summary, compressive strength, effective porosity and chloride ion content should be grouped into one cluster. For the analysis of the pore structure, one of the indicators can represent the others, which helps to reduce the interference among the indicators and provides a reference for subsequent studies; the rate of mass change, on the other hand, is grouped into a separate cluster, and its effect on the pore structure needs to be analysed separately.

## 4. Conclusions and Discussion

### 4.1. Conclusions

In order to investigate the law of basalt fibre to enhance the durability of concrete, this paper selects basalt fibre length as the main factor, supplemented by novel research methods such as neural networks, to study the rule of concrete resistance to multiple types of salt erosion. The salt was mixed in the experiments. The mass loss rate, effective porosity, compressive strength corrosion resistance factor and chloride ion content were selected to investigate the effects of basalt fibre length, curing age, erosion time and other influencing factors on the erosion resistance of basalt fibre fly-ash concrete. Additionally, the durability index of basalt fibre fly-ash concrete was predicted and analysed by eight artificial neural network models and a grey clustering analysis. The specific results are as follows:

(1) Increasing the length of basalt fibres can enhance the mechanical properties of concrete but weaken its durability. This is exacerbated by the mixing of fibres, but none of the increases are significant. The ranking of basalt fibre lengths for concrete resistance to mass loss and chloride erosion is as follows: 6 mm > 12 mm > 18 mm > 6 mm + 12 mm > 6 mm + 12 mm + 18 mm; the ranking of basalt fibre length for effective porosity is 6 mm + 12 mm + 18 mm > 6 mm + 12 mm > 18 mm > 12 mm > 6 mm.

(2) With large quantities of mineral admixtures and basalt fibres, the concrete can be eroded by the mixed salt solution for a relatively longer period of time. The 120 d erosion time is the turning point in the mechanical and durability performance of basalt fibre fly-ash concrete, compared to the usual 60 d for conventional concrete.

(3) The extension of the age of maintenance improves the performance of the concrete in terms of mechanical properties and durability, but to a lesser extent.

(4) The artificial neural network model was used to make predictions, and the results were accurate and reliable; the artificial neural network prediction model with decision trees had the highest accuracy with a score of 99.6. The decision tree was able to effectively model the durability-related indicators (effective porosity, mass loss rate, chloride ion content) of basalt fibre fly-ash concrete, providing a reference for subsequent studies such as simplified indoor tests.

(5) The grey clustering model analysis resulted in the compressive strength, effective porosity and chloride content being grouped into one cluster for uniform analysis, while the mass loss rate needed to be grouped separately. Analysis of the results reduces the collection of unnecessary variables (factors) and saves on the cost of the experiment.

*4.2. Discussion*

(1) Basalt fibre, fly ash and mineral powder are all green and pollution-free environmental protection materials, and their comprehensive application research in concrete is small, and it remains to be studied to develop different mixing ratios for different environments such as impact load, freeze–thaw environment and dry and wet cycle environment, for the selection of the best mix ratio.

(2) In this paper, basalt fibre fly-ash concrete is analyzed at the macroscopic level; the microscopic performance mechanism of basalt fibre dispersion and mineral admixture products, during the erosion period, on the mechanical properties and durability of concrete needs to be studied.

**Author Contributions:** This article was written by the following authors, who have contributed to this article below. S.S. (writing—original draft preparation, methodology); A.G. (validation, Conceptualization); R.L. (formal analysis, investigation) R.W. (data curation, resources); J.X. (data curation, project administration); F.W. (writing—review and editing, supervision); F.L. (writing—review and editing, supervision). All authors have read and agreed to the published version of the manuscript.

**Funding:** This work was supported by Scientific Research Fund project of Yunnan Education Department (grant number 2023J1974 and grant number 2023J1976) and the Yunnan University Professional Degree Graduate Student Practical Innovation Fund project (grant number ZC-22222374).

**Data Availability Statement:** Data available on request due to restrictions eg privacy or ethical. The data presented in this study are available on request from the corresponding author. The data are not publicly available due to [The data in this article has a certain value].

**Conflicts of Interest:** The authors declare no conflict of interest.

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
