# Peer review of "Study and Neural Network Analysis on Durability of Basalt Fibre Concrete"

_water, doi:10.3390/w15061016_

Round 1
Reviewer 1 Report
I recommend the paper for publication, however, there are some concerns, comments and suggestion should be addressed before publication:
1) There are grammar and typographic errors. Please correct these errors and further improve the language.
2) The novelty of this work must be presented in Conclusion section very clear.
3) For general readers, authors are encouraged to discuss other kind of related works structures such as: [(a) “Machine learning models for predicting the compressive strength of concrete containing nano silica”, Computers and Concrete, 30(1), 33-42.; (b) “Predicting elemental stiffness matrix of FG nanoplates using Gaussian Process Regression based surrogate model in framework of layerwise model”, Engineering Analysis with Boundary Elements, 143, 779-795.].
4) The results and figures are appropriate however; author should add more physical explanation for the observed results.
Author Response
请参阅附件

Reviewer 2 Report
The work involves an optimization study using artificial neural network approach of basalt fibers reinforcing concrete for applications related to the durability of the material against salt erosion. In my opinion the work is very interesting and useful for scholars and researchers working in the field of durability of cementitious materials.
There are a few points that should be addressed to improve the manuscript:
- Line 77: please specify the meaning of DIF
- Section 2.1: the materials used for the mix design should be described in more detail (type and strength class of the cement, type and granulometry of the aggregates, micromorphology of the basalt fibers and their elemental chemical composition)
- Sections 2.1.1 and 2.1.2: Do mechanical and durability tests follow specific standard methods? Please specify.
- Figure 1: Please correct "uotput" in the figure.
- Section 3 (Results and Discussions): Although the experimental results are well described, literature references to corroborate or contradict the obtained results are lacking. For example in Line 266, the authors state that mixed fibers tend to form agglomerates in the concrete, worsening the microstructural quality of the material and therefore encouraging its permeability. This evidence should be supported either by microanalyses or by findings in the literature.
- General comment: the effect of reinforcing fibers in cementitious materials is known to predominate on the flexural behaviour rather than compressive properties. Why didn't the authors also investigate this aspect in the optimization of basalt fibers?
Reviewer 3 Report
The manuscript, entitled "Study and neural network analysis on durability of basalt fiber concrete," presents an experimental study conducted on the durability of concrete with basalt fiber addition. However, the materials involved in the study were not described or analyzed, and many other issues must be addressed. The paper needs major revisions before it is processed further. Some comments follow:
Abstract: The abstract is written qualitatively. The majority of the qualitative statements ("large "small "increase in length," etc.) should be modified for quantified result comparisons. Also, the results obtained from the research should be presented quantitatively. This section must be suitable for separate or independent publication.
Introduction Section
The introduction should be significantly improved. Please conduct a comprehensive and exhaustive study of the previous literature (the latest research; currently, the references are old; there are no references from 2023 or 2022, and only a few from 2021—not enough). Please clearly highlight the pros and cons of previous results and justify the need for the current research.
Also, many affirmations have no background in the literature.Please provide corresponding citations or references to support the assertions from the first paragraph and lines 81–87.
2. Materials and Methods
Test mixes: What was the rationale for choosing this mixture? Why is this mixture relevant instead of others?
Please provide the main chemical and physical properties of all raw materials involved in the study. This data and information are mandatory to ensure experiment repeatability and reproducibility. Also, what was the rationale behind choosing the type and amount of fibers?
- Results and discussions
The microstructural analysis of the samples, as well as the evolution of the microstructure over time, should be of interest to the readers.Please provide such results.
Discussion section. The discussion section is missing. In the discussion section, a clear correspondence and comparison between the results of this study and those from the literature should be provided. Please improve.
Round 2
Reviewer 3 Report
Dear authors,
You have done a great job revising your manuscript. I have no further suggestions for you.
Best regards,